# Pre-Treatment Biomarkers of Anti-Tumour Necrosis Factor Therapy Response in Crohn’s Disease—A Systematic Review and Gene Ontology Analysis

**DOI:** 10.3390/cells8060515

**Published:** 2019-05-28

**Authors:** Boris Gole, Uroš Potočnik

**Affiliations:** 1Centre for Human Molecular Genetics and Pharmacogenomics, Faculty of Medicine, University of Maribor, Taborska ulica 8, SI-2000 Maribor, Slovenia; 2Laboratory for Biochemistry, Molecular biology and Genomics, Faculty of Chemistry and Chemical Engineering, University of Maribor, Smetanova ulica 17, SI-2000 Maribor, Slovenia

**Keywords:** Crohn’s disease, inflammatory bowel disease, anti-tumour necrosis factor, adalimumab, infliximab, certolizumab pegol, therapy response, biomarker

## Abstract

The most prominent treatment for the serious cases of Crohn’s disease (CD) are biological tumour necrosis factor (TNF) inhibitors. Unfortunately, therapy nonresponse is still a serious issue in ~1/3 of CD patients. Accurate prediction of responsiveness prior to therapy start would therefore be of great value. Clinical predictors have, however, proved insufficient. Here, we integrate genomic and expression data on potential pre-treatment biomarkers of anti-TNF nonresponse. We show that there is almost no overlap between genomic (annotated with tissue-specific expression quantitative trait loci data) and transcription (RNA and protein data) biomarkers. Furthermore, using interaction networks we demonstrate there is little direct interaction between the proposed biomarkers, though a majority do have common interactors connecting them into networks. Our gene ontology analysis shows that these networks have roles in apoptotic signalling, response to oxidative stress and inflammation pathways. We conclude that a more systematic approach with genome-wide search of genomic and expression biomarkers in the same patients is needed in future studies.

## 1. Introduction

Crohn’s disease (CD) is one of the two principal subtypes of Inflammatory Bowel Disease (IBD), a chronic autoimmune condition that may affect any part of the digestive system and seriously affects the quality of life [1,2,3]. The serious cases of CD/IBD are treated with biologicals, most prominently with tumour necrosis factor (TNF) inhibitors. Unfortunately, 20–40% of patients (in clinical studies) fail to respond to treatment while 23–46% of patients loose response over time [4,5]. Non-responsiveness to anti-TNF therapy in CD is therefore an important issue, both for patients who are exposed to potentially severe side-effects of the therapy that they do not benefit from, and for the healthcare systems that need to pay for the expensive yet inefficient treatment. With novel biologicals against alternative targets, such as ustekinumab (anti-IL12/23) and vedolizumab (anti-α_4_β_7_ integrin) for treatment of CD/IBD and thus possibility of greater stratification of the patients [6], pre-emptive prediction of non-response to anti-TNF is becoming even more meaningful.

Unfortunately, current clinical predictors of response to anti-TNFs do not satisfactorily predict the response. Genetic background and gene expression are therefore also under investigation [7,8,9,10,11,12,13]. There are several recent reviews on the topic, focusing either on clinical prediction of anti-TNF response [2,4,5,14,15] or summarizing genomic and/or expression data and proposing predictive sets of anti-TNF response markers [16,17,18,19,20]. Here, for the first time we integrate genomic and expression markers that could predict non-response to anti-TNF treatment in CD patients prior to treatment initiation into common interaction networks. We also use gene ontology (GO) tools to obtain some insight into signalling pathways/biological processes where future biomarkers of the anti-TNF therapy failure in the CD patients may hide.

## 2. Materials and Methods

### 2.1. Information Sources and Search, Study Selection

The PRISMA statement (http://www.prisma-statement.org/) was followed while conducting the present study. An electronic literature search of the PubMed was performed on November 30th, 2018. The search terms used were Crohn’s disease, Inflammatory Bowel Disease; anti-TNF, infliximab, adalimumab, golimumab, entanercept, certolizumab; response, non-response, outcome, therapy, treatment, prediction. Only publications in English language were eligible. Both U.K. and U.S. spellings were taken into account (i.e., tumour/tumor necrosis factor). The full search strategy including final query translation is described in Appendix A.

We included studies reporting on genetic and/or expression (RNA, protein) biomarkers predicting response to anti-TNF biologicals in adult CD patients measured at baseline (i.e., prior to therapy start). Study selection was performed in several rounds. Full selection process is described in Appendix A.

### 2.2. Data Collection and Integration

We collected data on genomic biomarkers and expression biomarkers measured in blood (peripheral blood mononuclear cells—PBMCs or serum) and colon (colon mucosa, stool) of CD patients prior to the anti-TNF therapy start. We collected all the relevant data regardless of the therapeutic used (infliximab, adalimumab, golimumab, unspecified) or how the clinical response was assessed (CDAI—Crohn’s Disease Activity Index, HBI—Harvey–Bradshaw Index, IBDQ—Inflammatory Bowel Disease Questionnaire, clinical assessment, etc.). Depending on the time-point of therapy response assessment we separated short-term response (week 2–22) and long-term response (>6 months).

As genomic markers we collected data on all the SNPs (single-nucleotide polymorphisms) which were connected to anti-TNF therapy response with a *p*-value < 0.05. The *p*-values corrected for multiple testing were also collected. If only uncorrected *p*-values were reported the adjusted *p*-values were calculated using Bonferroni correction—these adjusted *p*-values are marked with italics. To compare genomic markers to expression data only tissue-specific eQTL (expression Quantitative Trait Loci) data were acknowledged—Colon-sigmoid, Colon-Transverse and Small Intestine eQTLs to compare with colon mucosa expression data and blood eQTLs to compare to blood cells and serum expression data. The eQTL data were obtained from the GTEx Portal (https://www.gtexportal.org) on March 1st, 2019.

As expression biomarkers we collected the quantified data on RNA and proteins measured in anti-TNF therapy responders (Re) and nonresponders (NR) prior to therapy start. For mRNA individual qRT-PCR and microarray data were screened and biomarkers with log2 difference in expression level (NR to Re) of at least 1.5-fold collected; *p*-values adjusted for multiple testing <0.05 were defined as significant. From individual qRT-PCR experiments the ΔΔCt values were extracted. If only uncorrected *p*-values were reported the adjusted *p*-values were calculated using Bonferroni correction—these adjusted *p*-values are marked with italics. We downloaded microarray data from GEO database (https://www.ncbi.nlm.nih.gov/geo/). The log2-transformed fold differences in expression and the *p*-values were calculated with GEO2R available at GEO website based on LIMMA package. The *p*-values were adjusted with Benjamini–Hochberg method (False Discovery Rate). For RNA expression biomarkers in blood cells the uncorrected *p* < 0.05 was defined significant since none of the biomarkers met the adjusted *p* < 0.05 criterion. The probe annotations were determined using NetAffx Analysis Center (https://www.affymetrix.com/analysis) and BioMart (https://www.ensembl.org/biomart) services.

At protein level all quantifiable data (ELISA, immunohistochemistry, immunofluorescence, etc.) data were screened and biomarkers with >1.5-fold difference in expression level (NR to Re) extracted. The significance was defined at a *p*-value <0.05. For comparison of complex protein data with RNA and genomic data all the genes coding for the subunits of the protein complex were designated the same fold change and *p*-value.

### 2.3. Interactome Builds and Gene Ontology (GO) Analysis

Interactomes of the biomarkers were build with CytoScape platform (version 3.5.1, CytoScape Consortium, San Diego, CA, USA) based on protein–protein interactions downloaded from BioGrid database (version 3.5.170; https://thebiogrid.org) on March 6th, 2019. Besides interactions between biomarkers we also included the genes/proteins interacting directly with the biomarkers (interactors). To simplify the interactomes and avoid bias due to unequal numbers of interactors of individual biomarkers, the interactors connected to only single biomarkers were omitted from the network. GO analysis of biomarkers and of the whole interactomes was done within the CytoScape platform using the ClueGO plug-in (version 2.5.2, INSERM, Paris, France). Details on GO analysis settings are described in Appendix A.

## 3. Results

### 3.1. Literature Search

The electronic literature search yielded 9763 hits (Figure 1). The 9386 hits were either automatically excluded based on publication type (case reports, retracted publications, publications not reporting novel measurements, i.e., reviews, editorials, etc.) or excluded after manual screening for obviously irrelevant studies and studies not reporting data on adult CD patients (reporting experimental in vitro or animal data, data on ulcerative colitis, data on paediatric IBD). The remaining 377 publications were assessed for eligibility. The 301 publications not reporting on genomic or expression biomarkers, not comparing nonresponders to responders, not presenting separate data on CD patients, or reporting only expression data after therapy start were excluded. Additionally, we excluded nine more publications not reporting novel data (3× erratum, 1× study proposal, 2× commentary and 3× review not detected by automatic exclusion) and one where it was impossible to determine whether the data were collected pre- or post-therapy start. The remaining 66 publications (Appendix A) were included in the systematic review. Data on genomic and expression markers for GO analysis were extracted from 40 of these. The rest either reported only negative results, or the reported differences did not reach the *p*-value significance or fold change required for inclusion (see Methods).

### 3.2. Genomic Markers

We identified 40 studies trying to connect DNA sequence variability, i.e., single-nucleotide polymorphisms (SNPs) to anti-TNFs response in CD patients (Appendix A). Almost half of these studies (18/40) reported only negative results. The rest reported 72 SNPs connected to short- and 34 SNPs connected to long-term response to anti-TNF therapy in CD patients with a *p* < 0.05 (Appendix A, respectively). When the *p*-values are adjusted for multiple testing, 18 SNPs remain connected to short- and 3 SNPs connected to long-term response (Table 1). None of these SNPs has a confirmation in an independent patient cohort (without *p*-value adjustment four SNPs have independent confirmation—Appendix A). The SNP most significantly associated to both short- and long-term anti-TNFs response in CD patients after *p*-value adjustment is rs1130864, a 3′-UTR variant of the *CRP* gene (adjusted *p* = 4.09 × 10^−4^).

### 3.3. Expression Markers—RNA Level

Our search identified seven publications reporting RNA expression markers (4× on colon mucosa, 3× on PBMCs) of anti-TNF therapy response in CD patients (Appendix A). Four of them are based on (three different) genome-wide microarray data [12,30,31,32]. Additionally, Schmitt et al., 2018 [13] performed RNA-seq on colon mucosa samples, however only after the therapy start, thus these data were omitted from our search. Only one publication reported on long-term predictor of anti-TNF response [33]. In two of the publications, none of the data reached the fold change and/or significance level requested for inclusion in our analysis.

Looking at colon mucosa data, 89 baseline markers of short-term anti-TNF therapy response in CD patients were identified on RNA level (log2 fold change in expression between NR and Re ≤ −1.50 or ≥ +1.50; adjusted *p* < 0.05, Appendix A); all of them in the same microarray study [12]. Four were confirmed with an alternative method (qRT-PCR, Table 2) but not in an independent cohort of patients [30]. Expression of all but three RNA markers is higher in the NR compared to the Re patients. No baseline RNA predictors of long-term anti-TNF response in colon mucosa were reported thus far. RNA data from blood PBMCs identified five baseline markers of short-term and a single marker of long-term anti-TNF response in CD patients (Table 2). Expression of all but one of the short-term response markers is lower in NR compared to the Re patients, while the long-term response predictor is expressed higher in the NR. None of the blood RNA markers was independently confirmed.

### 3.4. Expression Markers—Protein Level

We identified 27 publications reporting baseline protein markers of anti-TNF response in CD patients (Appendix A). Majority (22 publications) investigated/reported serum markers and markers connected to short-term therapy response (23 publications). Seven studies reported protein markers in colon (four colon mucosa markers, three stool markers). None of the studies reported protein markers on PBMCs. Nine studies reported only negative results, and in an additional four studies the differences observed did not reach our inclusion criteria.

Only two protein markers measured in colon (colon mucosa or stool) fulfilled our inclusion criteria (fold change in expression between NR and Re ≤ 0.66 or ≥1.50; *p* < 0.05, Table 3) and both are connected to short-term therapy response. Calprotectin has multiple independent confirmations. Its baseline expression is higher in NR than in Re patients. Colon mucosa expression of TNF at baseline is lower in NR than in Re. In blood (serum) six protein markers were identified in connection to the short-term therapy response and two with long-term therapy response (Table 3). Among the short-term response markers, four have higher expression in NR than in Re patients at baseline (IL-8, IL-17A, TGF-β1 and TNF) while one (IL-15) has a lower expression. The results on the only short-term response serum marker with multiple independent confirmations (CRP) are ambiguous—its baseline expression was lower in NR patients than in Re patients in two studies [34,35] while in the other two studies [36,37] it was higher. Baseline expression of both long-term response serum markers was higher in NR compared to Re patients.

### 3.5. Integrated Data on Genomic and Expression Markers

Next, we integrated the collected data on genomic and expression (RNA and protein) markers of anti-TNF therapy response in CD patients. Genes connected to individual SNPs or corresponding to individual RNA and proteins were chosen as common denominators. Genomic markers (SNPs) were integrated via tissue-specific eQTL information. All the genes connected to specific SNPs via eQTL were designated the same *p*-values. RNA and protein names were directly translated into corresponding gene names. In case of complex proteins, all the subunits were taken into account with the same fold change and *p*-value as the original complex protein. In that way we created four lists of integrated data on anti-TNF response markers—for colonic short- and long-term response markers and for short- and long-term response blood markers (Appendix A).

Among integrated colonic markers of short-term response, nine have multiple independent confirmations—three of them (*FCGR2C*, *S100A8* and *S100A9*) on multiple levels of expression (Table 4). The other six (AC034220.3, *ACSL6*, *CASP9*, *HSPA7*, *SLC22A4* and *SLC22A5*) have multiple independent confirmations only on the level of DNA, either as individual SNPs connected (via eQTL) to single genes, multiple SNPs connected to the same gene or individual SNPs connected to multiple genes. In this integration, only one colonic marker of long-term response (*CCHCR1*) has an independent confirmation (two different SNPs corresponding to the same gene). Integration of blood markers resulted in four short-term response markers with multiple independent confirmations- three (AC116366.6, *RPS23P10* and *SLC22A5*) at the level of DNA and one (*CRP*) at the protein level (Table 4). None has confirmations on multiple levels of expression. The single blood marker of long-term response (*TREM1*) has confirmation of two levels of expression (RNA, protein), albeit from the same patient cohort [33].

### 3.6. Interactomes and Gene Ontology of Anti-TNF Therapy Response Markers

To check whether the identified response markers interact with each other and thus build common biological pathways we built interactomes of the integrated colonic and blood, short- and long-term response markers, using protein–protein interaction data available at BioGrid (see Methods). The markers which are not translated into proteins were omitted from this analysis. As it soon became clear that almost none of the identified response markers interact to each other (Figure 2 and Figure 3, Appendix A), we subsequently included also their “first neighbours” or “interactors”—proteins interacting directly to the response markers—which significantly improved the connectivity of the individual markers. Further, we used GO analysis to determine biological pathways that are common to the integrated response markers.

#### 3.6.1. Response Markers of the Colon

The 109 colonic short-term response markers form eight small networks of two to four individual markers (all together 14 marker-to-marker interactions), while the bulk (88) of these markers are not interacting directly (Appendix A). When also markers’ first interactors are included we get an interactome of 369 proteins with 655 interactions. The majority (77/109) of the markers and their interactors form a single network, while the rest remain separated from it (Appendix A). The 28 colonic short-term response markers remain disconnected from all other also in this broadened interactome. GO analysis of the colonic short-term response markers revealed 47 enriched GO terms (Appendix A). The most significant were cytokine activity (adjusted *p* < 5.43 × 10−5) and chemotaxis (adjusted *p* < 1.73 × 10^−4^). Analysis of the extended interactome revealed 429 enriched GO terms (Appendix A), with apoptotic process and regulation of response to stimulus as the most significant two (adjusted *p* < 1.84 × 10^−23^ and <7.52 × 10^−22^, respectively). Neither of the two GO analyses revealed any underrepresented GO terms.

We also built an interactome of colonic short-term response markers with multiple confirmations. Only two (*S100A8* and *S100A9*) of the nine markers interacted directly (Figure 2). Together with their first interactors, the extended interactome consists of 33 proteins with 53 interactions. The interactions here form three separate networks with one marker remaining completely disconnected. GO analysis of the markers alone finds three GO terms (Table 5) and analysis of the extended interactome 15 GO terms.

None of the 14 colonic long-term response markers interact with each other. When their interactors are included in the interactome (all together 42 proteins and 59 interactions) they form a single network, with five markers remaining completely unattached (Appendix A). GO analysis of long-term markers alone returned no results, while analysis of the extended interactome returned six enriched GO terms (Appendix A).

#### 3.6.2. Response Markers of the Blood

Short-term response markers identified in blood (40 proteins) form three small groups of two to three members (altogether four interactions) while a majority (33/40) of the markers do not interact (Figure 3). In the extended interactome including their first interactors (184 proteins, 329 interactions) almost all of the markers belong to the same network, with six markers unattached to it (Appendix A).

GO analysis of the blood short-term response markers returned 18 enriched GO terms (Table 6). The most significant ones are regulation of superoxide metabolic process (adjusted *p* < 2.48 × 10^−8^) and response to steroid hormone (adjusted *p* < 4.88 × 10^−8^). When the extended interactome was analysed, the result was enriched 249 GO terms (Appendix A). The two blood short-term response markers with multiple confirmations which can be translated into protein (*CRP*, *SLC22A5*) do not interact and also do not have common first interactors. GO analysis of the two terms also returned no results.

The 18 long-term response markers identified in blood do not interact directly. In the extended interactome including their first interactors (together 55 proteins, 82 interactions) they form a single network, with six markers remaining unattached to it (Appendix A). GO analysis of markers alone returned no results, while analysis of the extended interactome revealed 11 enriched GO terms (Appendix A).

## 4. Discussion

Genomic and expression markers of anti-TNF failure in CD patients are the topic of several recent reviews [16,17,18,19,20]. However, here, for the first time we merge these potential biomarkers from different levels of expression into common interaction networks and analyse the networks with gene ontology tools, aiming at finding common biological pathways behind the therapy non-response.

To identify the candidate genomic and expression predictors of anti-TNF response in CD we systematically screened the published data. We focused on markers that could predict response to therapy at baseline, prior to its start. In this regard, use of genomic markers seems the most logical one. DNA can be studied on easily obtainable material (i.e., blood, sputum, stool) prior to treatment, with no need for highly invasive procedures such as colonoscopy and recovery of colon mucosa samples. Additionally, DNA does not change over time or vary between tissues and cell types. We identified 40 studies trying to connect DNA sequence variability, i.e., SNPs to anti-TNF response in CD patients. Their success, however, was quite limited. So far, none of the SNPs proposed as genetic markers of anti-TNF response in CD reached the level of significance demanded for a genome-wide association study (*p* ≤ 5.0 × 10^–8^). Actually, only four SNPs, connected to short-term anti-TNF response in CD were confirmed in multiple independent patient cohorts, none, if *p*-values are adjusted for multiple testing. The reasons for such a poor output may be manifold. First, lack of reproducibility may be due to the limited selection of SNPs included in individual studies. Only three [10,11,48] looked at a larger array (>90.000) of loci, while the rest focused on a limited subset of genes, such as apoptotic genes [21], NFκB pathway [49], IBD-associated loci [7], or even just a couple of loci/SNPs. In fact, only ~10% of the SNPs were included in more than one study (data not shown). Next, most of the studies include relatively small numbers of cases (typically 100–300 patients, Appendix A) and thus lack statistical power needed to reach the high significance levels demanded for a genome-wide association [50]. To amend this, probably a big multi-centre effort similar to the one finding the 163 IBD susceptibility loci [1] is needed.

Unlike in genomic studies, the time-point of data collection is important in expression studies. Since therapy itself changes the expression profiles in patients [51,52], and our focus were pre-treatment predictors of therapy response, we excluded all the data collected post anti-TNF treatment induction. Contrary to the genomic studies, RNA expression studies of anti-TNF response markers in CD are few, but several are genome-wide [12,13,31,32,52,53] and thus less biased than currently published genomic studies. Surprisingly, despite genome-wide screens, there are no independently confirmed baseline RNA markers of anti-TNF response in CD patients. In fact, we had to lower our originally intended criteria for data inclusion (log2 FC NR/Re from ≥2.00 to ≥1.50) in order to get any hits at all with the PBMCs. Additionally, we had to lower also significance limit for PBMCs (*p* < 0.05 instead of adjusted *p* < 0.05).

Studies reporting protein markers are more numerous than the RNA studies, though most report only data on a few serum proteins (CRP, albumin, haemoglobin) or faecal calprotectin. Some measured several (7–12) serum proteins [41,42,43,45,54] but so far there are no published baseline proteome-wide studies on the anti-TNF response markers. Many reported results are negative or the differences observed are marginal and would not allow for a robust cut-off for practical use. The majority of the protein markers also lack independent confirmation. The two protein markers with multiple independent confirmations are serum CRP and faecal calprotectin. Both are nowadays readily measured in CD/IBD patients to assess therapy effectiveness post-induction and are being assessed also as baseline response predictors [37]. While high baseline faecal calprotectin seems to be predictive of therapy failure, there is so far no unified cut-off value for practical use. The proposed cut-offs vary widely, from 160µg/g to 863µg/g [37,38,39]. On the other hand, data on baseline serum CRP are so far inconclusive; although several studies confirm differential baseline CRP expression between NR and Re patients, the direction of this difference varies between studies (see Table 3, [20]).

Altogether, as with SNPs, poor reproducibility is a problem also with expression biomarkers of anti-TNF response in CD. Additionally, as with SNPs, lack of “whole genome” studies (at least at protein level) and low numbers of included patients (RNA level, Appendix A) may be reasons for this. Additionally, gene expression reflects the processes in the patient and is influenced by environmental factors. For example, patient cohorts from selected studies include anti-TNF naïve patients, patients already switched from another anti-TNF drug, and practically all of them had additional therapies prior and/or concomitant to anti-TNF induction.

A possible solution for poor reproducibility of results on individual levels of expression (DNA, RNA, protein) is integration of data from different levels. However, it also poses an additional problem, namely, how to connect data on SNPs with expression data. For the purpose of this review, genes connected to individual SNPs or corresponding to individual RNA/proteins were chosen as common denominators. We used publicly available tissue-specific eQTL data to annotate SNPs to specific genes. In several cases that meant that the connection is not the same as was intended in the original reference. For example, many studies focused on SNPs in or near the TNF gene [22,27,34,55,56,57] however current eQTL data do not confirm connection between these SNPs and the *TNF* gene. Additionally, while for example rs10210302 is clearly connected to *ATG16L1* expression in skin, thyroid or subcutaneous adipose tissue, there is no eQTL data connecting the SNP to the *ATG16L1* expression in colon or blood. On the other hand, it turned out that several SNPs that were detected as potential biomarkers in only single studies influence the same gene and thus do have some sort of independent confirmation (Table 4). Another entity difficult to compare to other levels of expression, complex proteins, we dissected to their subunits (for example calprotectin to S100-A8 and S100-A9) and designated the fold changes and *p*-values of the whole protein to all the subunits. Using this approach, out of identified 108 genomic, 95 RNA and 10 protein markers we created a selection of 15 markers with multiple confirmations. Although the total number of baseline markers with multiple confirmations more than doubled compared to the individual level comparisons, it is difficult not to observe that there is no repeatability between genomic and expression markers. Only *FCGR2C* was detected both at genomic and expression (RNA) level.

To determine whether the identified markers at least belong to the same biological processes and pathways we build interactomes (extra for colon and blood markers, short- and long-term response markers) based on publicly available protein–protein interaction data and performed GO analysis of these interactomes. Similar approaches were used for IBD in general to build interactomes of genes annotated from SNPs without use of tissue-specific eQTL data [11], baseline blood expression markers [32] and post-baseline colon expression markers [52] connected to anti-TNF response. In our CD-oriented study, it soon became clear that there are almost no direct interactions between the identified response markers, so we introduced into the networks also proteins, with which the markers directly interact. Such approach can lead to better insight into complex connections when limited data on direct interactions between proteins of interest are available [58]. In that way, almost all markers for short-term response are connected into common networks of interactions. On the contrary, only 2/3 of the long-term response markers are connected in common networks. This is reflected also in the GO analysis. Analysis of both marker sets (colon, blood) returned no results, while analysis of the extended interactomes resulted in very unspecific terms connected to degradation of faulty proteins, response to UV light and cytoskeleton organization.

The most significantly enriched GO terms for the short-term response markers in colon were connected with cytokine activity and immune response, chemotaxis and leukocyte migration- in general inflammation. The GO analysis of blood short-term response markers resulted in enriched GO terms connected to ROS (Reactive Oxygen Species) response, steroid hormone response and immune response. Looking at both extended interactomes (with direct interactors of markers) apoptosis—response to reactive oxygen species (ROS)—and general terms linked to signalling- and protein-metabolism are also enriched.

While the increased colon inflammation despite anti-TNF treatment is quite literally the definition of non-response, pre-treatment disease severity in general is not predictable of anti-TNF response in CD [4]. This in a way is counterintuitive, as faecal calprotectin, a marker of inflammation in the gut [59] is under consideration as a baseline therapy response predictor, as already mentioned above. Since our GO analyses show calprotectin (*S100A8/9*) connection to several GO terms not directly associated to inflammation (see Table 5) it is possible that predictive value of calprotectin originates from one of them, as is discussed below. Differences in cytokine signalling were suggested also by Gaujoux et al., 2018 [32] based on their GO analysis of baseline blood RNA markers. However, several studies measuring whole arrays of serum cytokines on protein levels found differences between NR and Re only in individual interleukins (IL-8 [41], IL-15 [42], IL-17A [43], TNF [45]). A recent study that measured baseline protein levels of several cytokines in colon mucosa also found no differences [13]. Connection between anti-TNF loss of response and ROS response was also suggested in a post-treatment study of expression markers in colon [60]; however, measurements of the actual ROS in baseline serum of the NR and Re patients did not find any difference between the two groups [61]. It is possible that the “potential” differences in inflammation, cytokine and ROS responses between the NR and Re patients become relevant only after the therapy start. Indeed, the majority of the ROS response associated genes identified in our analysis (*TNF*, *TGFB1*, *PARK7*, *LTBR*) are also among the genes connected to apoptotic process, which according to our GO analyses is strongly associated with non-response to anti-TNF in both colon and blood (Appendix A, individual genes not shown). Apoptosis is at the centre of so far the most comprehensive (post-treatment induction) model of anti-TNF non-response in CD [13]. In the model, failure of the CD4+ T-cells to undergo apoptosis leads to increased inflammation, meaning that the potential baseline difference in inflammation response as suggested by our GO analysis becomes relevant. Among the key players of the Schmitt et al. model (TNF, TNFR2, STAT3, IL23A, IL23R) [13] our screen for baseline therapy response predictors detected only TNF, further underlining that sets of baseline response predictors and post-therapy response markers can differ significantly despite belonging to the same signalling pathways (GO terms). The anti-apoptotic effect on T-cells in the model is augmented by macrophages; and regulation of macrophage activation via *FCGR2C* (low affinity IgG receptor) and calprotectin (*S100A8/9*) is indeed enhanced in NR already at baseline, as suggested by our GO analysis of the colon biomarkers with multiple independent confirmations (Table 5, Figure 1).

## 5. Conclusions

In conclusion, currently genomic markers of anti-TNF response in CD patients are not reaching sufficient significance levels and there is practically no reproducibility between genomic and baseline expression markers. To amend this, whole-genomes transcriptomes and/or proteomes should be measured in the same individuals. We and others [62] are already creating appropriate biobanks of patients’ samples; however, multinational efforts to create larger patient cohorts will be needed if we are to reach also the sufficient statistical power of data. For now, GO analysis indicates that pre-treatment differences in apoptosis, ROS and inflammation response between nonresponders and responders to anti-TNF therapy be should thoroughly researched for potential baseline biomarkers.

## Figures and Tables

**Figure 1 cells-08-00515-f001:**
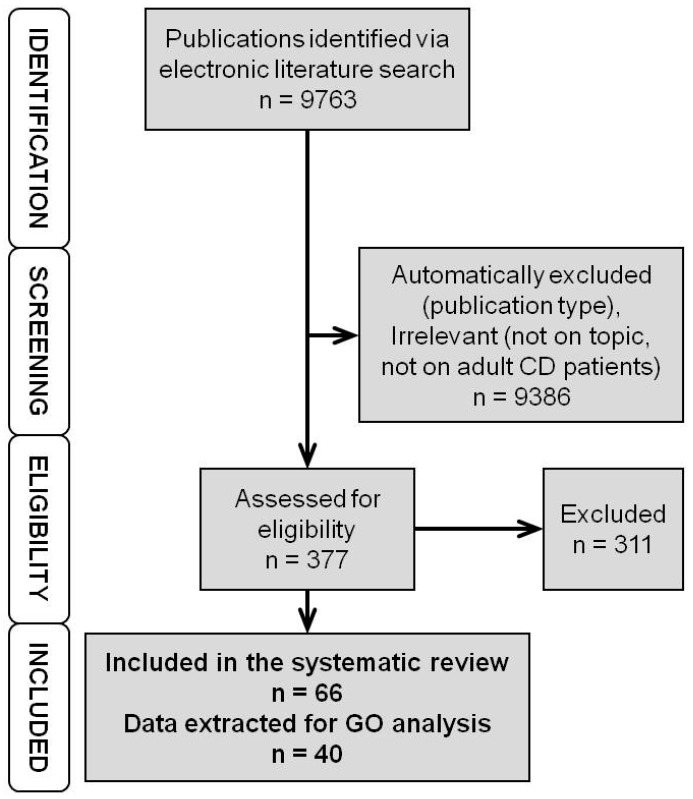
Flowchart of the literature search and selection. First, we identified the publications via electronic literature search. Then, using automatic screening, we excluded publication types not reporting new measurements/data and with manual screening irrelevant studies. The remaining publications were assessed for eligibility. Those reporting suitable (quantifiable) data on genomic or expression markers of anti-TNF response measured prior to therapy start were included in the systematic review. (TNF – Tumour Necrosis Factor, CD – Crohn’s Disease, GO – Gene Ontology).

**Figure 2 cells-08-00515-f002:**
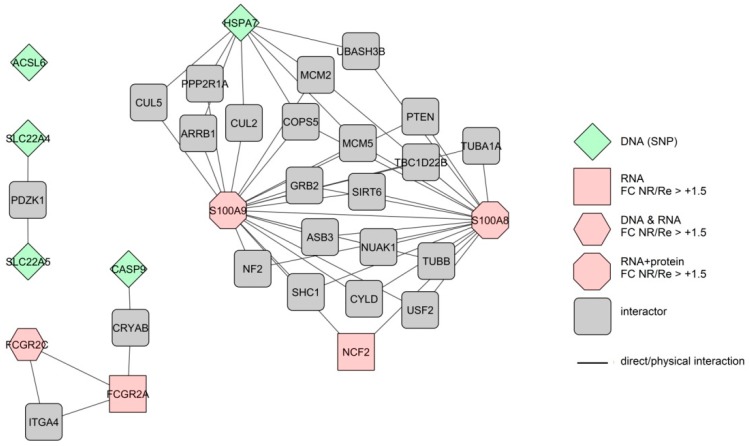
Extended interactome of the independently confirmed, integrated colonic markers linked to the short-term anti-TNF therapy response in CD patients. Different forms represent markers measured at different levels of expression. The colours represent differences in expression level (FC—fold change) between therapy nonresponders (NR) and responders (Re). Lines represent direct protein–protein interactions.

**Figure 3 cells-08-00515-f003:**
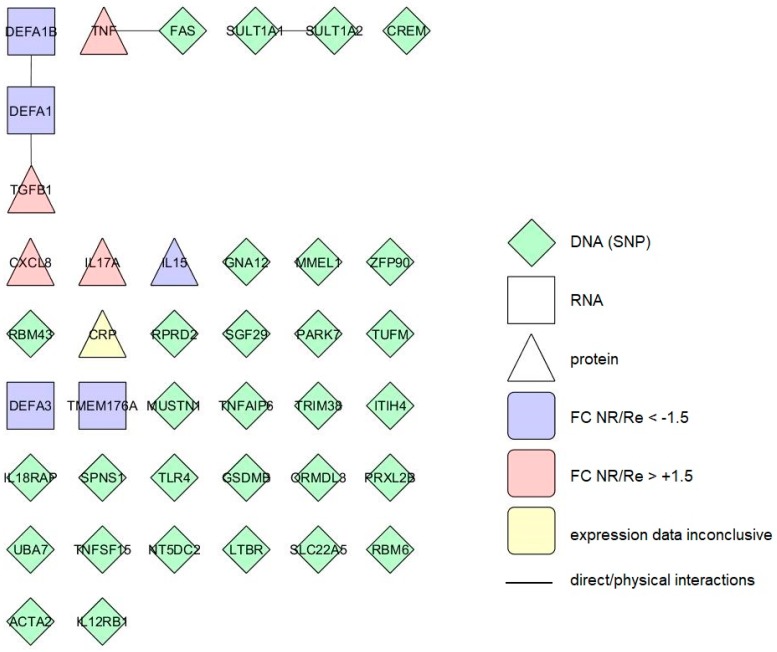
Interactome of the integrated blood markers linked to the short-term anti-TNF therapy response in CD patients. Different forms represent markers measured at different levels of expression. The colours represent differences in expression level (FC—fold change) between therapy nonresponders (NR) and responders (Re). Lines represent direct protein–protein interactions.

**Table 1 cells-08-00515-t001:** Single-Nucleotide Polymorphisms (SNPs) linked to anti-TNF response in CD patients with adjusted *p* < 0.05.

SNP	*p*-Value	Adjusted *p*-Value	Allele/Genotype Association		Reference
***short-term response***					
rs1130864	4.09 × 10^−5^	*4.09 × 10^−4^*	C	Re	[8]
rs763110	N.D.	1.00 × 10^−3^	CC/CT	Re	[21]
rs12469362	N.D.	1.30 × 10^−3^	T	Re	[22]
rs10495565	N.D.	1.40 × 10^−3^	G	Re	[22]
rs1056204	N.D.	1.40 × 10^−3^	C	Re	[22]
rs4464248	N.D.	1.80 × 10^−3^	G	Re	[22]
rs396991	3.00 × 10^−3^	3.00 × 10^−3^	GG	Re	[23]
rs1813443	2.00 × 10^−3^	6.00 × 10^−3^	CC	NR	[24]
rs9373839	1.13 × 10^−3^	*1.13 × 10^−2^*	C	Re	[8]
rs2071303	8.20 × 10^−3^	*1.64 × 10^−2^*	G	Re	[9]
rs1061624	5.00 × 10^−3^	*2.00 × 10^−2^*	A	NR	[25]
rs1568885	7.00 × 10^−3^	2.10 × 10^−2^	TT	NR	[24]
rs976881	2.00 × 10^−3^	*2.40 × 10^−2^*	A	NR	[26]
rs10210302	8.10 × 10^−4^	2.67 × 10^−2^	CC	Re	[7]
rs1143634	N.D.	2.70 × 10^−2^	C	NR	[27]
rs4645983	N.D.	3.00 × 10^−2^	TT	Re	[21]
rs2522057	2.40 × 10^−2^	*4.80 × 10^−2^*	C	NR	[28]
rs35260072	2.40 × 10^−2^	*4.80 × 10^−2^*	G	NR	[28]
***long-term response***					
rs1130864	4.09 × 10^−5^	4.09 × 10^−4^	C	Re	[8]
rs9373839	1.13 × 10^−3^	*1.13 × 10^−2^*	C	Re	[8]
rs1799724	N.D.	4.00 × 10^−2^	CC	Re	[29]

Adjusted *p*-values not originally reported, but calculated here are represented in italics. NR (allele/genotype frequency higher in nonresponsive patients), Re (allele/genotype frequency higher in responsive patients); N.D. (not defined).

**Table 2 cells-08-00515-t002:** RNA markers linked to anti-TNF response in CD patients.

Gene Name	NCBI Gene ID	log2 FC NR/Re	*p*-Value	Adjusted *p*-Value	Reference
***colon (colon mucosa)**^1^*					
*short-term response*					
*IL13RA2*	3598	2.761	1.27 × 10^−5^	2.37 × 10^−2^	[12]
		2.184	<1.00 × 10^−2^	*<5.00 × 10^−2^*	[30]
*IL11*	3589	3.306	1.41 × 10^−5^	2.37 × 10^−2^	[12]
		3.184	<1.00 × 10^−2^	*<5.00 × 10^−2^*	[30]
*STC1*	6781	1.514	7.49 × 10^−5^	3.89 × 10^−2^	[12]
		2.396	<1.00 × 10^−2^	*<5.00 × 10^−2^*	[30]
*PTGS2*	5743	2.206	8.55 × 10^−5^	3.94 × 10^−2^	[12]
		3.059	<1.00 × 10^−2^	*<5.00 × 10^−2^*	[30]
***blood (PBMCs)***					
*short-term response*					
DEFA1	1667	−1.885	6.72 × 10^−3^	9.13 × 10^−1^	[32]
DEFA3	1668	−1.885	6.72 × 10^−3^	9.13 × 10^−1^	[32]
DEFA1B	728358	−1.950	6.84 × 10^−3^	9.13 × 10^−1^	[32]
SCARNA4	677771	1.701	2.24 × 10^−2^	8.96 × 10^−1^	[31]
TMEM176A	55365	−1.712	2.36 × 10^−2^	8.96 × 10^−1^	[31]
*long-term response*					
TREM1	54210	2.341	2.00 × 10^−2^	2.00 × 10^−2^	[33]

Adjusted *p*-values not originally reported, but calculated here are represented in italics. ^1^ Only markers confirmed by two independent methods are listed. For the full list of colonic RNA markers of response see Appendix A.

**Table 3 cells-08-00515-t003:** Protein markers linked to anti-TNF response in CD patients [38,39,40,41,42,43,44,45,46].

Protein Name	FC NR/Re	*p*-Value	Reference
***colon (colon mucosa, stool)***			
*short-term response*			
*colon mucosa*			
TNF	0.366	4.00 × 10^−4^	[40]
*stool*			
**calprotectin**	3.053	<5.00 × 10^−3^	[38]
	13.889	<5.00 × 10^−3^	[39]
	4.463	3.00 × 10^−2^	[37]
***blood (serum)***			
*short-term response*			
TGF-β1	1.724	<5.00 × 10^−3^	[44]
**CRP**	0.571	2.00 × 10^−2^	[34]
	0.568	7.00 × 10^−3^	[35]
	2.732	1.50 × 10^−2^	[36]
	4.563	4.00 × 10^−2^	[37]
IL-8 (CXCL8)	1.786	1.00 × 10^−2^	[41]
IL-15	0.054	1.00 × 10^−2^	[42]
TNF	14.286	3.50 × 10^−2^	[45]
IL-17A	19.231	4.00 × 10^−2^	[43]
*long-term response*			
TREM1	2.096	1.00 × 10^−3^	[33]
IL-6	1.961	2.90 × 10^−2^	[46]

Protein markers with more than one independent confirmation are represented in bold.

**Table 4 cells-08-00515-t004:** Independently confirmed integrated markers linked to anti-TNF response in CD patients.

Gene Name	NCBI Gene ID	SNP	*p*-Value	Adjusted *p*-Value	Data Type	Reference
***Colon (colon-sigmoid/colon-transverse/small intestine eQTL, colon mucosa/stool expression)**^1^*						
*short-term response*						
AC034220.3	N.A.	rs2522057	2.40 × 10^−2^	4.80 × 10^−2^	DNA	[28]
		rs35260072	2.40 × 10^−2^	4.80 × 10^−2^		
		rs1050152	2.40 × 10^−2^	7.92 × 10^−1^		[7]
		rs2631372	2.60 × 10^−2^	8.58 × 10^−1^		
ACSL6	23305	rs2522057	2.40 × 10^−2^	4.80 × 10^−2^	DNA	[28]
		rs35260072	2.40 × 10^−2^	4.80 × 10^−2^		
		rs1050152	2.40 × 10^−2^	7.92 × 10^−1^		[7]
CASP9	842	rs4645983	N.D.	3.00 × 10^−2^	DNA	[21]
			7.20 × 10^−3^	2.38 × 10^−1^		[7]
FCGR2C	9103	rs396991	3.00 × 10^−3^	3.00 × 10^−3^	DNA	[23]
		N.A.	1.34 × 10^−4^	4.55 × 10^−2^	RNA	[12]
		rs396991	2.10 × 10^−2^	6.93 × 10^−1^	DNA	[7]
HSPA7	3311	rs396991	3.00 × 10^−3^	3.00 × 10^−3^	DNA	[23]
			2.10 × 10^−2^	6.93 × 10^−1^		[7]
S100A8	6279	N.A.	7.67 × 10^−6^	2.37 × 10^−2^	RNA	[12]
			<5.00 × 10^−3^	N.D.	protein	[38]
			<5.00 × 10^−3^	N.D.		[39]
			3.00 × 10^−2^	N.D.		[37]
S100A9	6280	N.A.	5.61 × 10^−5^	3.57 × 10^−2^	RNA	[12]
			<5.00 × 10^−3^	N.D.	protein	[38]
			<5.00 × 10^−3^	N.D.		[39]
			3.00 × 10^−2^	N.D.		[37]
SLC22A4	6583	rs35260072	2.40 × 10^−2^	4.80 × 10^−2^	DNA	[28]
		rs1050152	2.40 × 10^−2^	7.92 × 10^−1^		[7]
SLC22A5	6584	rs35260072	2.40 × 10^−2^	4.80 × 10^−2^	DNA	[28]
		rs1050152	2.40 × 10^−2^	7.92 × 10^−1^		[7]
		rs2631372	2.60 × 10^−2^	8.58 × 10^−1^		
long-term response						
CCHCR1	54535	rs1800629	4.90 × 10^−2^	3.43 × 10^−1^	DNA	[47]
		rs1799724	2.50 × 10^−3^	4.00 × 10^−2^		[29]
***blood (blood eQTL, serum/PBMCs expression)***						
*short-term response*						
AC116366.6	N.A.	rs2522057	2.40 × 10^−2^	4.80 × 10^−2^	DNA	[28]
		rs35260072	2.40 × 10^−2^	4.80 × 10^−2^		
		rs1050152	2.40 × 10^−2^	7.92 × 10^−1^		[7]
		rs2631372	2.60 × 10^−2^	8.58 × 10^−1^		
CRP	1401	N.A.	2.00 × 10^−2^	N.D.	protein	[34]
			7.00 × 10^−4^			[35]
			1.50 × 10^−2^			[36]
			4.00 × 10^−2^			[37]
RPS23P10	100419471	rs396991	3.00 × 10^−3^	3.00 × 10^−3^	DNA	[23]
			2.10 × 10^−2^	6.93 × 10^−1^		[7]
SLC22A5	6584	rs2522057	2.40 × 10^−2^	4.80 × 10^−2^	DNA	[28]
		rs35260072	2.40 × 10^−2^	4.80 × 10^−2^		
		rs1050152	2.40 × 10^−2^	7.92 × 10^−1^		[7]
*long-term response*						
TREM1	54210	N.A.	2.00 × 10^−2^	2.00 × 10^−2^	RNA	[33]
			1.00 × 10^−3^	1.00 × 10^−3^	protein	

Adjusted *p*-values not originally reported, but calculated here are represented in italics. N.A. (not applicable), N.D. (not defined).

**Table 5 cells-08-00515-t005:** GO analysis of the independently confirmed, integrated colonic markers linked to the short-term anti-TNF therapy response in CD patients.

GO ID	GO Term	Ontology Source	*p*-Value	Adjusted *p*-Value	% Assoc. Genes	Genes
***GO analysis of the markers***						
GO:0015909	long-chain fatty acid transport	BP	4.28 × 10^−6^	4.28 × 10^−6^	3.53	*ACSL6, S100A8, S100A9*
GO:0030888	regulation of B cell proliferation	BP	3.84 × 10^−6^	7.69 × 10^−6^	3.66	*FCGR2C, S100A8, S100A9*
GO:0043030	regulation of macrophage activation	BP	2.70 × 10^−6^	8.11 × 10^−6^	4.11	*FCGR2C, S100A8, S100A9*
***GO analysis of the extended interactome***						
GO:0015838	amino-acid betaine transport	BP	1.09 × 10^−7^	1.53 × 10^−6^	60.00	*PDZK1, SLC22A4, SLC22A5*
GO:0015879	carnitine transport	BP	1.09 × 10^−7^	1.53 × 10^−6^	60.00	*PDZK1, SLC22A4, SLC22A5*
GO:0015697	quaternary ammonium group transport	BP	3.81 × 10^−7^	4.95 × 10^−6^	42.86	*PDZK1, SLC22A4, SLC22A5*
GO:0015695	organic cation transport	BP	3.92 × 10^−6^	4.70 × 10^−5^	21.43	*PDZK1, SLC22A4, SLC22A5*
GO:0030165	PDZ domain binding	MF	8.88 × 10^−6^	9.76 × 10^−5^	6.90	*PDZK1, PTEN, SLC22A4, SLC22A5*
GO:0072337	modified amino-acid transport	BP	1.42 × 10^−5^	1.42 × 10^−4^	14.29	*PDZK1, SLC22A4, SLC22A5*
GO:0015696	ammonium transport	BP	3.84 × 10^−5^	3.46 × 10^−4^	10.34	*PDZK1, SLC22A4, SLC22A5*
GO:0008180	COP9 signalosome	CC	1.88 × 10^−4^	5.64 × 10^−4^	6.12	*COPS5, GRB2, HSPA7*
GO:0042770	signal transduction in response to DNA damage	BP	7.67 × 10^−5^	6.13 × 10^−4^	4.00	*CASP9, GRB2, S100A8, S100A9*
GO:0001540	amyloid-beta binding	MF	9.47 × 10^−5^	6.63 × 10^−4^	7.69	*CRYAB, FCGR2A, FCGR2C*
GO:0006898	receptor-mediated endocytosis	BP	1.88 × 10^−4^	7.51 × 10^−4^	3.17	*ARRB1, FCGR2A, FCGR2C, GRB2*
GO:0090342	regulation of cell aging	BP	1.36 × 10^−4^	8.17 × 10^−4^	6.82	*NUAK1, S100A8, S100A9*
GO:2001235	positive regulation of apoptotic signalling pathway	BP	1.71 × 10^−4^	8.55 × 10^−4^	3.25	*CYLD, PTEN, S100A8, S100A9*
GO:0007569	cell aging	BP	1.12 × 10^−3^	1.12 × 10^−3^	3.33	*NUAK1, S100A8, S100A9*
GO:0008630	intrinsic apoptotic signalling pathway in response to DNA damage	BP	6.61 × 10^−4^	1.32 × 10^−3^	4.00	*CASP9, S100A8, S100A9*

MF (molecular function), BP (biological process), CC (cellular component).

**Table 6 cells-08-00515-t006:** GO analysis of the integrated blood markers linked to the short-term anti-TNF therapy response in CD patients.

GO ID	GO Term	Ontology Source	*p*-Value	Adjusted *p*-Value	% Assoc. Genes	Genes
***GO Analysis of the Markers***						
GO:0090322	regulation of superoxide metabolic process	BP	1.38 × 10^−9^	2.48 × 10^−8^	22.73	*CRP, LTBR, PARK7, TGFB1, TNF*
GO:0048545	response to steroid hormone	BP	3.05 × 10^−9^	4.88 × 10^−8^	4.94	*DEFA1, DEFA1B, DEFA3, GNA12, PARK7, RBM6, TGFB1, TNF*
GO:0071396	cellular response to lipid	BP	2.97 × 10^−9^	5.04 × 10^−8^	3.01	*CXCL8, DEFA1, DEFA1B, DEFA3, GNA12, PARK7, RBM6, TGFB1, TLR4, TNF*
GO:2000379	positive regulation of reactive oxygen species metabolic process	BP	2.46 × 10^−7^	3.70 × 10^−6^	8.47	*CRP, LTBR, PARK7, TGFB1, TNF*
GO:0051701	interaction with host	BP	4.37 × 10^−7^	6.12 × 10^−6^	4.76	*CXCL8, DEFA1, DEFA1B, SLC22A5, TGFB1, TRIM38*
GO:0002227	innate immune response in mucosa	BP	5.89 × 10^−7^	7.66 × 10^−6^	13.79	*DEFA1, DEFA1B, DEFA3, RBM6*
GO:0071383	cellular response to steroid hormone stimulus	BP	8.14 × 10^−7^	9.77 × 10^−6^	4.29	*DEFA1, DEFA1B, DEFA3, GNA12, PARK7, RBM6*
GO:0030518	intracellular steroid hormone receptor signalling pathway	BP	5.05 × 10^−6^	5.56 × 10^−5^	4.63	*DEFA1, DEFA1B, DEFA3, PARK7, RBM6*
GO:0019731	antibacterial humoral response	BP	6.98 × 10^−6^	6.98 × 10^−5^	7.55	*DEFA1, DEFA1B, DEFA3, RBM6*
GO:0032930	positive regulation of superoxide anion generation	BP	7.99 × 10^−6^	7.19 × 10^−5^	17.65	*CRP, LTBR, TGFB1*
GO:0031640	killing of cells of other organism	BP	1.31 × 10^−5^	1.05 × 10^−4^	6.45	*DEFA1, DEFA1B, DEFA3, RBM6*
GO:0061844	antimicrobial humoral immune response mediated by antimicrobial peptide	BP	2.38 × 10^−5^	1.66 × 10^−4^	5.56	*DEFA1, DEFA1B, DEFA3, RBM6*
GO:0035821	modification of morphology or physiology of other organism	BP	3.02 × 10^−5^	1.81 × 10^−4^	3.21	*DEFA1, DEFA1B, DEFA3, RBM6, TGFB1*
GO:0044003	modification by symbiont of host morphology or physiology	BP	3.78 × 10^−5^	1.89 × 10^−4^	10.71	*DEFA1, DEFA1B, DEFA3, RBM6, TGFB1*
GO:0032677	regulation of interleukin-8 production	BP	7.08 × 10^−5^	2.83 × 10^−4^	4.21	*DEFA1, DEFA1B, TGFB1*
GO:0071260	cellular response to mechanical stimulus	BP	1.81 × 10^−4^	3.63 × 10^−4^	6.38	*DEFA1, DEFA1B, TGFB1*
GO:0071222	cellular response to lipopolysaccharide	BP	1.44 × 10^−4^	4.31 × 10^−4^	3.51	*CRP, PARK7, TLR4, TNF*
GO:1901224	positive regulation of NIK/NF-kappaB signalling	BP	4.33 × 10^−4^	4.33 × 10^−4^	4.76	*FAS, LTBR, TLR4*

BP (biological process).

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
