# Peer review of "Pre-Treatment Biomarkers of Anti-Tumour Necrosis Factor Therapy Response in Crohn’s Disease—A Systematic Review and Gene Ontology Analysis"

_cells, 2019, doi:10.3390/cells8060515_

Round 1
Reviewer 1 Report
The systematic review “Pre-treatment biomarkers of anti-Tumor necrosis factor therapy response in Crohn’s disease – a systematic review and gene ontology analysis” by Gole & Potočnik (Cells-496443) has the relevant aim to verify, by the analysis of published studies, whether biomarkers of response to anti-TNF therapy could be identify.
In the paper, the introduction presents the matter clearly. Results and discussion need to be improved.
Specific comments
Page 4, lines 144-147: the two SNPs described as most significantly associated with short term (rs704191) or long term (rs9904253) anti-TNF response are not among those that remained connected to response after p-values adjustment. This create confusion in interpreting the results. Similarly, the four SNPs connected to short term response confirmed in more than one independent studies, are included in table 1 (significant adjusted p-values), but present in suppl. Table 2. I suggest to organize better the presentation of these results. Moreover, at least in table 1, I suggest to add a column reporting the response-associated alleles.
Page 5, lines 171-173 and table 2: The statement “all the short term response markers is lower in NR compared to Re patients” is not supported by the log2 FC NR/Re value of SCARNA4. Please verify data in table 2.
Table 3: I suggest to distinguish TNF from calprotectin, the first from colon while the second from stool. Moreover, considering the wide use of fecal calprotectin some more details on the cut-off that might distinguish Re from NR could be of interest and practical use. The same consideration might be applied to CRP.
In the discussion, the authors should focus on the most significant GO terms by reporting the potential role of the associated genes. In other words, is there any experimental evidence available supporting a pathogenetic role for the specific genes?
Author Response
Comment 1:
Page 4, lines 144-147: the two SNPs described as most significantly associated with short term (rs704191) or long term (rs9904253) anti-TNF response are not among those that remained connected to response after p-values adjustment. This create confusion in interpreting the results. Similarly, the four SNPs connected to short term response confirmed in more than one independent studies, are included in table 1 (significant adjusted p-values), but present in suppl. Table 2. I suggest to organize better the presentation of these results. Moreover, at least in table 1, I suggest to add a column reporting the response-associated alleles.
Response 1:
We reorganized the part on genomic markers to make it clearer and put more emphasis on the data with adjusted p-values (see page 4, lines 133-142). In line with that, we omitted the specific mentioning of the SNPs most significantly associated with anti-TNF response without p-value adjustment. These data remain accessible in the Supplementary Tables 2 and 3, which were reorganized, so that the SNPs are ordered by p-values making it easier to find the most significant hits.
As suggested we also added data on response-associated alleles/genotypes (depending on which the authors reported) to both Table 1 (page 5) and Supplementary Tables 2 and 3.
Comment 2:
Page 5, lines 171-173 and table 2: The statement “all the short term response markers is lower in NR compared to Re patients” is not supported by the log2 FC NR/Re value of SCARNA4. Please verify data in table 2.
Response 2:
We thank the reviewer for noticing this detail. The data in Table 2 are correct. Two words were erroneously omitted when preparing the final version of the manuscript for submission. The proper text (now in page 5, lines 163-164) is “Expression of all but one of the short-term response markers is lower in NR compared to the Re patients...”
Comment 3:
Table 3: I suggest to distinguish TNF from calprotectin, the first from colon while the second from stool. Moreover, considering the wide use of fecal calprotectin some more details on the cut-off that might distinguish Re from NR could be of interest and practical use. The same consideration might be applied to CRP.
Response 3:
We now clearly separated the colon mucosa and stool markers in Table 3 (page 7).
Regarding possibility to distinguish Re from NR at baseline using faecal calprotectin we added a sentence to the discussion presenting the currently proposed cut-off values (pages 14, lines 333-335). Regarding CRP, so far the results on baseline CRP levels are ambiguous, showing opposing results (higher baseline CRP expression in nonresponders in two studies, lower expression in other two studies- see the results section, page 6, lines 185-188 and discussion, page 14, lines 335-338) and thus inconclusive.
Comment 4:
In the discussion, the authors should focus on the most significant GO terms by reporting the potential role of the associated genes. In other words, is there any experimental evidence available supporting a pathogenetic role for the specific genes?
Response 4:
In the discussion, we added some more details on genes connected to the most significant/interesting GO terms (see page 15, lines 388-418), which, together with the information already included before should sufficiently explain current experimental support for our GO analysis. Please note that the supplementation of the discussion demanded additional reference, hence numbering of references 59-87 changed to 60-88, which we therefore changed also in Supplementary tables 1-3, 6 and 7.
Reviewer 2 Report
This is a state of the art manuscript exploring predictors of response to Anti-TNF in Crohn’s disease in a systematic review of 66 different manuscripts and, when feasible, also their raw data. The topic is of utmost importance with the advent of many novel biologics as diverse therapeutic options. I am not a content expert on Gene Ontology and gene expression methods and thus I cannot comment on the core methodology of the manuscript (it all seems clear and comprehensive to me but a content expert should review). The systematic approach to data collection and selection of outcomes, the use of PRISMA reporting guidelines, the clear presentation of results with standardized definitions are significant strengths of this robust manuscript. But the most novel and useful aspect of the paper is the attempt to integrate all findings in a comprehensive approach. I suggest accepting the manuscript “as is” except for one little suggestion to entirely delete the first paragraph of the introduction; readers of this manuscript do not need the student-like description of the disease. Very well done.
Author Response
Comment 1:
I suggest accepting the manuscript “as is” except for one little suggestion to entirely delete the first paragraph of the introduction; readers of this manuscript do not need the student-like description of the disease. Very well done.
Response 1:
We agree that the first paragraph of the introduction is probably unnecessary for the majority of the readers. We reduced the introductory description of the disease to two and a half lines and merged it with the next paragraph. Consequently, we also slightly reorganized the first sentence of the former second paragraph (see page 1, lines 27-30).